# Health impacts of industrial mining on surrounding communities: Local perspectives from three sub-Saharan African countries

**Andrea Leuenberger**[1,2]*, **Mirko S. Winkler**[1,2], **Olga Cambaco**[3], **Herminio Cossa**[1,2,3], **Fadhila Kihwele**[4], **Isaac Lyatuu**[1,2,4], **Hyacinthe R. Zabré**[1,2,5], **Andrea Farnham**[1,2], **Eusebio Macete**[3], **Khátia Munguambe**[3,6]

1 Swiss Tropical and Public Health Institute, Basel, Switzerland, 2 University of Basel, Basel, Switzerland, 3 Manhiça Health Research Centre, Maputo, Mozambique, 4 Ifakara Health Institute, Dar es Salaam, Tanzania, 5 Research Institute of Health Sciences, Ouagadougou, Burkina Faso, 6 Faculty of Medicine, Eduardo Mondlane University, Maputo, Mozambique

* andrea.leuenberger@swisstph.ch

**Data Availability Statement:** The data used for this paper are of qualitative nature (i.e. verbatim transcript of focus group discussions). Participants

## Abstract

Industrial mining projects can play an important role in global sustainable development if associated health risks are minimised and opportunities maximised. While a broad body of evidence from quantitative studies exists that establishes the interlinkages between mining operations and effects on public health, little research has been conducted investigating health impacts from the perspective of affected communities. This is particularly true in sub-Saharan Africa, where about a third of the remaining global mineral resources are endowed and health-related indicators for sustainable development are lagging behind. In this multi-country qualitative study, we explore community perceptions regarding impacts of industrial mining on their health and well-being. In nine study sites in Burkina Faso, Mozambique and Tanzania, we conducted 83 participatory focus group discussions with a total of 791 participants (385 men, 406 women). Our findings reveal a broad range of perceived impacts on environmental, economic and social determinants of health, with secondary health implications related to morbidity, mortality and well-being. Overall, perceived negative impacts prevailed, mainly related to environmental pollution, change in livelihoods or social disruption. Perceived positive impacts on health and well-being were related to interventions implemented by the mines such as new or improved water sources, health care facilities, roads and schools. The consistency of these findings across countries and study sites suggests a structural problem and indicates a pressing need to address health by acting on the wider determinants of health in mining regions. Participatory health impact assessment should be strengthened in host countries to foster strategic interventions, include marginalised population groups, and protect and promote the health of local communities. By including community perspectives on health before and during project implementation, policymakers can take advantage of economic opportunities while avoiding the pitfalls, bringing their communities closer to achieving good health and well-being goals by 2030 and beyond.

**Funding:** This work was supported by the Swiss Programme for Research on Global Issues for Development (r4d Programme, http://www.r4d.ch/), which is a joint funding initiative by the Swiss Agency for Development and Cooperation (SDC) and the Swiss National Science Foundation (SNSF). The grant (No. 169461) was awarded to the principle investigator of the research project framing this study (MW). The funders had no role in study design, data collection and analysis, decision to publish, or preparation of the manuscript.

**Competing interests:** The authors have declared that no competing interests exist.

# Introduction

Africa endows about a third of global reserves of natural resource of metals and minerals [1]. The continent is currently hosting about 700 active industrial mines and many more are planned, partly in response to the global transition toward a low-carbon future [1, 2]. Large-scale extraction projects are important drivers for the economic development of low- and middle-income countries and thus, can play a critical role in the frame of the 2030 Agenda for Sustainable Development [3, 4]. This also encompasses implications for health, which is a central aspect of sustainable development [5, 6]. For example, if mining projects work in partnership with local health systems, it is expected that better health and well-being of local communities can be achieved in communities living in mining regions [7, 8]. It has been shown that upgraded infrastructure and increased socio-economic conditions can result in a diverse set of health opportunities such as reduced childhood mortality, improved public infrastructure and increased wealth index [9–11].

On the other hand, potential health benefits of mining industries are opposed by potential negative effects on the environment, society and health, which need to be adequately managed to strive toward sustainable development [12]. Indeed, a large body of research investigates environmental pollution in mining areas, including related consequences for the health of affected populations [13–15]. Project-induced in-migration has also been identified as a major concern in mining regions that is associated with an array of health risks, including overburdened public infrastructure and social services or increased burden of sexually transmitted diseases [16–18].

Overall, existing literature focusing on health and well-being of communities affected by large natural resource extraction projects is predominately quantitative in nature and mostly focused on specific diseases or specific causes for health impacts [19]. Only few studies have focused on the wider determinants of health in the context of resource extraction [20]. In addition, there is a paucity of qualitative studies investigating health as multi-factorial issue [21]. Consequently, voices from affected communities tend to be heavily underrepresented in the scientific literature at hand. Hence, the missed opportunity to capture perceptions and attitudes in a comprehensive manner is problematic and limits full engagement with communities.

Focus group discussions (FGDs), based on fruitful discussions about views and experiences shared with others, can be particularly insightful to gain understanding of community perspectives on health and well-being [22]. A community-based approach might generate new findings and potentially reveal the "underlying truth", compared to quantitative studies, which are often hypothesis-driven [23]. Additionally, well-designed qualitative studies in multiple countries have been shown to preserve the richness of data and simultaneously reveal country specific aspects [24, 25]. The inductive nature of qualitative research allows the investigation of complex issues, such as health, and particularly offers insights into "how" health and well-being is impacted by virtue of social processes [26, 27]. Despite the potential of qualitative community-based research from different countries, comparative research investigating the perceived health and well-being of local communities affected by natural resource extraction projects is currently missing.

To address this gap, we aimed to explore (i) community member's perception of the mechanisms through which their health has been affected by mining projects and (ii) to identify the implications of different extraction projects for health and well-being in Burkina Faso, Mozambique and Tanzania. To focus on the perception of affected communities, the following four research questions guided the current research: (i) how has health and well-being been impacted by mining projects? (ii) which determinants of health have played a role for the

above mentioned relationship? (iii) what are the differences and similarities of perceived health impacts of mining projects in affected communities in Burkina Faso, Mozambique and Tanzania? (iv) what are site-specific implications for health and well-being in the communities surrounding different extraction projects?

## Methods

This study was conducted as part of a mixed methods and multi-country research project, aiming to promote Health Impact Assessment (HIA) practice in sub-Saharan Africa [28, 29]. Quantitative data from existing routine health data (e.g. Demographic and Health Survey, DHS) are being used to better understand impacts of natural resource extraction projects on population health by comparing impacted and comparison sites. Qualitative data were collected in the four project partner countries (Burkina Faso, Ghana, Mozambique and Tanzania) to explore the perspectives of different stakeholders, including the voices of affected communities. The generated evidence will be used to inform a policy dialogue toward the institutionalisation of HIA.

### Study sites

The work presented included qualitative data from nine study sites in three sub-Saharan African countries, namely Burkina Faso, Mozambique and Tanzania (Fig 1). Although the study was also conducted in the fourth partner country of the framing project (Ghana), data was incomplete at the time point of the analysis and thus, not included in the current work. In all study countries, environmental impact assessments (EIA) are required by law to obtain a mining license. Yet, specific national regulations for health impact assessment and the rigorous inclusion of health in other forms of impact assessment remain an unmet need [30, 31]. Characteristics of the study sites investigated are compiled in Table 1. Key aspects of the different countries and sites are described in the following paragraphs.

In Burkina Faso the study was carried out in communities around three gold mines, namely the Bissa Gold Mine, the Yaramoko Gold Mine and the Houndé Gold Mine. The Bissa Gold Mine is located in the north of the country, next to a small city and surrounded by three villages, where data was collected. The mine opened in 2013 and is an open-pit mine. Repeated terrorist attacks happened in the region, but they were not related to the mining activities. The Yaramoko Gold Mine is located in the western part of the country, between Ougagadougou (capital) and Bobo-Dioulasso (second largest city). Data was collected in two neighbourhoods of the closest town that were near the mine. The mine is operational since 2016 and extracts the ore from underground. Less than 100 km away from the Yaramoko Gold Mine, one can find the Houndé Gold Mine. The open-pit mine opened in 2017. Three villages located next to the mine were selected for data collection.

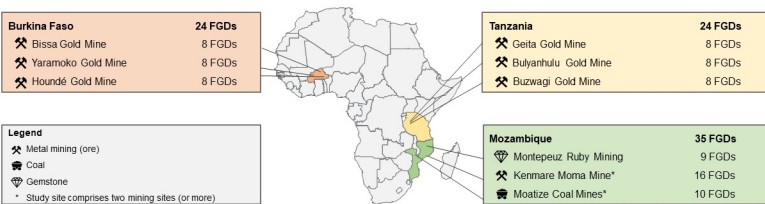

**Fig 1. Overview of countries and study sites.** Map indicating study countries, study sites and number of focus group discussions (FGDs) conducted (between March 2019 and January 2020).

**Table 1. Characteristics of study sites.**

| Mining project | Operator (location of corporate office) | Resource extracted | Year opened | Type of mine | Language | Time point of data collection | Number of FGDs | Additional information |
|---|---|---|---|---|---|---|---|---|
| **Burkina Faso** | | | | | | | | |
| Bissa Gold Mine | Nord Gold SE (United Kingdom) | Gold | 2013 | Open-pit | Mooré | May 2019 | 8 | • Repeated terrorist attacks in the region (but not primarily related to the mine) |
| Yaramoko Gold Mine | Roxgold Inc (Canada) | Gold | 2016 | Underground | Bwamou, Dioula, Mooré | April 2019 | 8 | *n/a* |
| Houndé Gold Mine | Endeavour Mining (United Kingdom) | Gold | 2017 | Open-pit | Bwamou, Dioula, Mooré | March 2019 | 8 | *n/a* |
| **Mozambique** | | | | | | | | |
| Montepuez Ruby Mining | Gemfields (Mozambique) | Ruby | 2011 | Open-pit | Emakuwa | May 2019 | 9 | • Province marked by insurgencies |
| Kenmare Moma Mining | Kenmare Resources Plc (Ireland) | Titanium | 2007 and expected start of extraction in 2020 | Open-pit | Emakuwa, Ekoti | July 2019 | 16 | • Study site comprised two mining sites (8 FGDs conducted around each mining site, 16 in total) |
| | | | | | | | | • One mining site was under construction during data collection |
| | | | | | | | | • Mines located next to the coast |
| Moatize Coal Mines | ICVL (India) and Vale (Brazil) | Coal | 2005 (ICVL) 2013 (Vale) | Open-pit | Nyungwe | September 2019, January 2020 | 10 | • Several mines located in the same study area |
| | | | | | | | | • Villages investigated were primarily impacted by the ICVL and Vale mine |
| | | | | | | | | • Mines and communities located next to the Zambezi and Rovuboe rivers |
| **Tanzania** | | | | | | | | |
| Bulyanhulu Gold Mine | Acacia Mining Plc (United Kingdom) | Gold | 2001 | Underground | Swahili | April, May 2019 | 8 | • Reduced production (no extraction of raw material) |
| Buzwagi Gold Mine | Acacia Mining Plc (United Kingdom) | Gold | 2009 | Open-pit | Swahili | April 2019 | 8 | • Reduced production (no extraction of raw material) |
| Geita Gold Mine | Anglo Gold Ashanti (South Africa) | Gold | 2000 | Open-pit | Swahili | March, April 2019 | 8 | *n/a* |

Overview of characteristics of study sites, including information about the mining projects and data collection (FGD: focus group discussion, ICVL: International Coal Venture Private Limited).

In Mozambique, the following three study sites were selected for our study: the Montepuez Ruby Mining, the Kenmare Moma Mining with its two mining sites and two mining sites of the Moatize Coal Mines. All are open-pit mines, located in the northern part of the country. The Montepuez Ruby Mining is located in a region currently affected by armed insurgency. The mine opened in 2011 and is placed between four communities where data was collected. The Kenmare Moma Mining has two Titanium minerals mining sites located along the coast. The first mining site was opened in 2004 and data has been collected in the four villages closest to the mine. The second mining site has been established a few kilometres south and was under construction during the data collection. Another four villages, which were heavily impacted by the implementation of the mine, were selected for data collection. In both mines,

a magnetic plant is used for processing the raw material. The complex of the Moatize Coal Mines is next to Tete, the closest city and in proximity to the Zambezi and Rovuboe rivers. In the study area, several mining pits of different commodities (e.g. gold, graphite) are operated by different companies. The 14 communities included in the study were primarily affected by coal mines operated by Vale Mozambique and International Coal Venture Private Limited (ICVL). The mining site of Vale Mozambique opened in 2005 and ICVL in 2013.

In Tanzania data was collected around three gold mines, namely, Bulyanhulu Gold Mine, Buzwagi Gold Mine and Geita Gold Mine. All three gold mines are located in the north-west of the country and not more than 100 km away from Lake Victoria. The Bulyanhulu mine opened in 2001 and was in a reduced production state at the time point of data collection, meaning that raw material was processed but not extracted. Data were collected in three communities in proximity to the Bulyanhulu mine. Near the Buzwagi mine, which is 6 km away from Kahama town, three communities were selected for our study. The mine has been operating since 2009 but was in a reduced production state at the time point of the field research. Around the Geita Gold Mine, which is next to Geita, the main city of the district, another three villages were selected for data collection. This open-pit mine opened in the year 2000 and was fully operational at the time point of visit.

## Terminology

Hereafter, we use the general term "community" to refer to different country-specific terminologies for the unit investigated in the field, such as villages, settlements, neighbourhoods or localities. Similarly, we use the term "health facility" in its broadest sense to refer to any locations where health care is provided and therefore embodies the different levels of health care (i.e. primary, secondary and tertiary) and country-specific terminologies (e.g. centres for health and social promotion, dispensaries, health centres, clinics or hospitals). "Perceived health impacts" refer to the changes induced by the industrial mining project on the wider determinants of health as reported by the study participants. The health determinants include a broad range of factors, including water, income generating activities and health care among others [32, 33]. "Perceived health outcomes" are reported secondary consequences for the health and well-being of affected communities caused by the changes in health determinants.

## Ethical approval

Ethical approval for the study was received in all three countries prior to beginning the fieldwork. Hence, the study was approved by national and institutional review boards in the three countries as well as Switzerland. Namely, the study protocol was approved by the Ethics committee for health sciences (Comité d'Ethique pour la Recherche en Santé) in Burkina Faso (No. 2019–013), and Institutional Committee on Bioethics in Health at the Manhiça Health Research Centre (Comité Institucional de Bioética para Saúde do Centro de Investigação em Saúde de Manhiça) in Mozambique (No. CIBS-CISM/048/2018), Ifakara Health Institute Review Board (No. 32–2018) and National Institute for Medical Research (NIMR) in Tanzania (No. 2969), and the Institutional Review Board of the Swiss Tropical and Public Health Institute (Swiss TPH), the Ethics committee of Northwestern and Central Switzerland (Ethikkommission Nordwest- und Zentralschweiz, EKNZ) in Switzerland (No. 2018–00386). All study participants provided written informed consent to participate in the study and to be recorded during the FGDs. Participants were reimbursed for travel expenses or provided with snacks and refreshments in accordance to local research standards.

## Sampling and recruitment

In each study site, the field teams from the partnering health research institute collaborated closely with local gatekeepers, including healthcare professionals and local government officers. To obtain an overview of the study site, a transect walk was conducted under the guidance of the gatekeepers. This approach allowed the research teams to identify eligible communities in a systematic manner. Once in the selected communities and with the assistance of the respective community leaders or community health workers, FGDs were held with purposively sampled community members. In order to generate comparative data from the different settings and countries, two to four sessions were planned with 6–10 participants per session. People above 18 years and knowledgeable about the community and its dynamic were invited to take part in gender-separated sessions.

## Data collection

Data were collected from March 2019 (first FGD in Burkina Faso) until January 2020 (last FGD in Mozambique). With the help of a project specific data collection manual, the same participatory data collection tool was administered in all study sites. The tool guide was translated from English to official languages (i.e. French in Burkina Faso, Swahili in Tanzania and Portuguese in Mozambique) by the local researchers involved in the same project, who were, thus, knowledgeable about the purpose of the research and the site's context. The core of the discussion was the participatory listing of perceived impacts on the wider determinants of health, followed by a categorization and ranking to further stimulate discussion. An example of the outcome is given as (see S1 Fig). All FGDs were conducted in local languages (Bwamou, Dioula and Mooré in Burkina Faso; Emakuwa and Nyungwe in Mozambique; Swahili in Tanzania), moderated by trained research assistants, and held in central places of the community (e.g. community offices, schools, guesthouses). An additional observer from the field team assisted the moderator during the sessions. The sessions were audio-recorded and lasted on average 81 minutes (Burkina Faso: 90 minutes; Mozambique: 68 minutes; Tanzania: 91 minutes). Demographic background information of all participants were registered in standardised forms prior to each discussion. All fieldwork activities were coordinated and supervised by PhD candidates involved in the research project. Occasionally, these activities were overseen by their local supervisors. Regular exchange among PhD students in the field ensured consistency across the countries.

## Data management and analysis

All FGDs were transcribed by research assistants into official languages (i.e. French, English, and Portuguese) of the three countries. All of them were reviewed for quality control by local researchers involved in the project [34].

To explore the community members' perception of the mechanisms through which their health is affected by mining projects and identify unique aspects of study countries and settings, a thematic analysis was conducted. Multiple authors were involved in the analytical process. The coding was mainly done by HC, OC and AL, using Nvivo (Nvivo 12 Pro, QSR International), and closely supervised by a senior social behavioural scientist (KM). The analysis began with repeated review of a few transcripts to develop a code system. After coding together a first transcript, coding for another transcript was done individually, followed by a discussion. The coding system was based on predefined categories informed by the data collection tool as well as themes emerging from the data. Rooted in the concept of the wider determinants of health, perceived impacts were categorised into (i) environmental aspects, (ii) economic aspects and (iii) aspects of the social services and organisation. Emerging sub-codes

were added during the analysis (see S1 Table). While analysing transcripts from the different countries in parallel, the coding system became static when about one third of all the data was coded. During the entire coding process, the researchers collaborated closely and discussed unclassified segments, most interesting findings and tentative interpretation on a regular basis. The interpretative examination during the regular discussion meetings enabled the international research team to reveal similarities but also differences across the countries and settings, as well as scrutinise the pathway of perceived impacts on health and the interconnection between different impacts. A separate code for cases that were unclear was used in case of doubt and to ensure interrater-reliability. Additionally, illustrative quotations were captured in a separate code. The findings were visualized based on the coding queries administered in the qualitative data analysis software (NVivo).

## Results

### Study population

An overview of the socio-demographic background of study participants per country is given in Table 2. In total, 791 community members (406 women, 385 men) participated in our study, with an average of nine community members per FGD. In Burkina Faso and Tanzania, eight FGDs were conducted in each of the three study sites (24 FGDs per country in total), while in Mozambique 8–16 were conducted in each of its three sites (35 FGDs in total; see Fig 1). Participants were aged between 18 and 89 years, and on average 42 years old, while they had lived on average for 20 years in their community. Participants attended on average less than four years of formal education and were mainly active in the agricultural sector. Only few participants reported to be employed by the mining companies (*n* = 13). In all mining sites,

**Table 2. Study participants.** Number of focus group discussion (FGD) study participants and their socio-demographic characteristics by country and in total.

| | Burkina Faso (24 FGDs) | Mozambique (35 FGDs) | Tanzania (24 FGDs) | Total (83 FGDs) |
|---|---|---|---|---|
| **Number of participants** | | | | |
| Men | 115 (49.8%) | 181 (48.0%) | 89 (48.6%) | 385 (48.7%) |
| Women | 116 (50.2%) | 196 (52.0%) | 94 (51.4%) | 406 (51.3%) |
| **Total** | **231** | **377** | **183** | **791** |
| **Average number (and range) of participants per FGD** | | | | |
| Men | 9.6 (6–10) | 10.6 (6–12) | 7.6 (6–8) | 9 (6–12) |
| Women | 9.6 (7–10) | 10.8 (8–13) | 7.8 (6–10) | 9.2 (6–13) |
| **Total** | **9.6 (6–10)** | **10.7 (6–13)** | **7.6 (6–10)** | **9.1 (6–13)** |
| **Average age (and age range) in years** | | | | |
| Men | 42.0 (23–71) | 44.9 (19–89) | 47.6 (19–77) | 44.7 (19–89) |
| Women | 30.8 (18–49) | 43.8 (29–83) | 41.5 (20–77) | 39.4 (18–83) |
| **Total** | **36.7 (18–71)** | **44.4 (19–89)** | **44.5 (19–77)** | **42.0 (18–89)** |
| **Average years (and range) living in the community** | | | | |
| Men | 26.2 (3–67) | 36.9 (3–89) | 22.1 (2–66) | 30.3 (2–89) |
| Women | 13.2 (1–44) | 36.4 (1–83) | 18.9 (1–77) | 25.4 (1–83) |
| **Total** | **19.7 (1–67)** | **36.7 (1–89)** | **20.5 (1–77)** | **27.8 (1–89)** |
| **Average number (and range) of years of school attended** | | | | |
| Men | 2.7[a] (0–10) | 3.8 (0–12) | 7.4 (0–14) | 4.4 (0–14) |
| Women | 1.2[a] (0–10) | 1.4 (0–12) | 7.3 (0–14) | 2.9 (0–14) |
| **Total** | **1.9[a] (0–10)** | **2.6 (0–12)** | **7.4 (0–14)** | **3.6 (0–14)** |

[a] Data from two FGDs with men and two FGDs with women missing (totally from four FGDs missing).

artisanal mining or the search for abandoned coal mining pits was reported during the FGDs as common income generating activity for local communities.

## Perceived health impacts and related health outcomes

In all three countries, a wide diversity of perceived impacts of the industrial mines on wider determinants of health were revealed during the FGDs. Codes about the various impacts and a description of the themes that emerged, complemented with indicative quotations, are compiled and added as (see S1 Table). Perceived changes in health status, resulting from changes in health determinants, emerged as secondary effects that can be clustered in three overarching themes, namely: (i) morbidity (including diseases, disabilities and injuries), (ii) mortality and (iii) well-being. Exemplary quotations for these different themes are compiled in Table 3 (quotations in the original languages are given in S2 Table). An overview of the distribution of perceived positive and negative impacts and related health outcomes are shown in Fig 2. Fig 3 indicates the proportion of FGDs having discussed the different topics of impacts, stratified by countries and type of resource extracted. In the following sections, we describe participants' perceptions of different impacts on the wider determinants of health and related implications for their health and well-being, including country-specific examples.

## Perceived health impacts related to health determinants

**Environmental determinants.**   With regards to environmental determinants of health, participants reported impacts related to water, air, housing and living environment, soil and land (incl. agriculture) and sanitation and hygiene.

Most prominently, participants reported water quality and availability of reliable water sources. Community members complained about pollution of different water bodies by the chemicals used in the mine as well as completely dried up wells due to the increased water demand related to the mining activities. Around gold and coal mines, they also perceived that the rain water is polluted with particles or chemicals coming from the mine, which were washed from the roof in their water collection system. Several communities around gold mines in Tanzania reported water flowing from the mine to water sources, be it during normal circumstances or in the case of a tailing dam bursting:

*Here is a dam for keeping dirty water containing chemicals. Something happened and water flowed to outside the mining and contaminated water in a pond near the mining then flowed to the rice farms. People who were planting rice got skin infection and animals died because of drinking the contaminated water*

*TZ1_L2*

Linked with the contamination and scarcity of water, participants from all countries reported different water-borne diseases as well as skin rashes after being exposed. They revealed an increase in malformations in newborns or miscarriages, which they linked to chemicals in the water. On the other hand, in almost all study sites participants felt relieved by interventions upgrading the water infrastructure. Many new water pumps or taps were installed by the mining companies. According to the participants, the availability of water contributed to their health and well-being.

Secondly, different types of air pollution were described during the FGDs. Most importantly, respondents referred to dust coming from the mine and polluting the environment. Of note, heavy dust emissions were particularly reported around gold and coal mines where mining-related blasting (explosions) were common. Around gold mines dust was described to

**Table 3. Exemplary quotations of coded health outcomes.**

| | Positive | Negative |
|---|---|---|
| **Morbidity** | | |
| Diseases | • We put the soap next to the toilet so that one can wash the hand after defecation in order to **avoid the diseases** (BF2_L2) | • Since the implementation of the mine **we are falling sick** (BF2_L7) |
| | • We appreciate the bed net they gave us because it **reduces the mosquito bites** a little (MZ2.2_L1) | • The company causes **many diseases** (MZ2.2_L1) |
| | • We acknowledge because we can go to the hospital [. . .] to **receive medicines** from [the company], they give it and we take it (MZ2.2_L7) | • We were using unprotected wells, they had no much effect but after the mining [. . .], now there are **many diseases** and we think it is caused by water (TZ2_L7) |
| | • Now due to the presence of a dispensary children are taken there for testing and if they **diagnosed** with malaria they are given medicines (TZ3_L2) | • Dust is produced when they are grinding stones and this dust is spreading to the community where people are living. As a result people are getting **cough** (TZ2_L3) |
| | • There is **testing** of HIV frequently in our community (TZ2_L3) | • Also there is an increase of HIV **transmission** because many people from different regions came to work in the mining (TZ2_L1) |
| Accidents and injuries | *n/a* | • The traffic **accidents**, which we encounter as part of vehicles and their trucks (BF1_L4) |
| | | • [A car] from the Kenmare company **almost hit me** (MZ2.1_1) |
| | | • If adult caught for stealing Magwangala [ore in the mine] are **beaten** so badly because you are matured enough to know what you are doing is wrong (TZ3_L7) |
| Disabilities | • We have good cooperation with the NREP [natural resource extraction project] we are allowed to employ this woman who is a **cripple** (TZ1_L6) | • They came to search everything with their machines causing quiet geniuses starting to mix with humans this caused that certain women gave birth to **weird children** (BF1_L3) |
| | • The mining sponsors for medical treatment for children who have midomo ya sungura [**open-cleft**] (TZ3_L3) | • These diseases now are full and it is different from the past, the topic of «**rheumatism**» is created because of the dust (MZ2.1_L2) |
| | | • I was working in the mining but later they discovered that I have **sight problem** (TZ1_L3) |
| **Mortality** | | |
| | • We appreciate the mosquito nets that they gave us [. . .], without these **nets**, nobody would survive here in the community (MZ2.2_L11) | • Those who are suffering from the risk to die in their houses because they don't have anything to eat anymore (BF1_L2) |
| | • We were travelling long distance to **get health services** so it was costful also we losted our beloved ones a mother or child because of delivering on the way to the health facility and some women decided to deliver at home because they couldn't afford the cost to go to the health center all these led to maternal deaths in our community but now situation has been improved (TZ1_L2) | • Since the mine has come, honestly there is no health anymore and the people **are not living long** anymore (BF3_L1) |
| | | • There are many diseases that **kill us** (MZ2.2_L2) |
| | | • Those who have the courage to go in the bush [for artisanal mining], even if **they can be killed** because they try to dig (MZ3_L7) |
| | | • When the houses have cracks the vibrations may cause the house to fall down and **you will all die** in there (TZ3_L3) |
| | | • Going to steal remaining of the processed stones called "magwangala", so **the security guard kills them** for self-defence (TZ3_L1) |
| | | • The use of unsafe water is what causes **miscarriage** (TZ2_L2) |
| | | • If you consider all these factors the answer is people from [our village] are **the living dead** (they are expecting to die) because they are using things which are not safe (TZ3_L7) |

(*Continued*)

**Table 3.** (Continued)

| | Positive | Negative |
|---|---|---|
| **Well-being** | | |
| | • The people can look after their children compared to before, because some parents who are working in the mine (BF3_L5) | • We **don't have a good quality of life** anymore, it is not easy to live permanently «under the dust» (BF2_L8) |
| | • Drilled well has improved **healthy** because water is safe (TZ1_L6) | • There is no water, no jobs, it is the **misery** (BF1_L2) |
| | • The NREP [natural resource extraction project] has constructed school, classrooms and toilets which has improved our **well-being** (TZ2_L5) | • When they are blasting, we are **frightened** any moment (BF1_L4) |
| | • So when mining built this health center here we **feel good**, we get treatment near our homes and we don't waste transport fee as we used to do before (TZ2_L7) | • We eat badly, **we don't sleep comfortably**, our farms have reduced to 50 meters and the rest are for these white people (MZ2.2_L3) |
| | | • People are currently living with **fear** because boundaries between the community and mining are not clear (TZ1_L5) |
| | | • We **do not trust** the water that we use (TZ1_L5) |
| | | • We are in a very **difficult situation** because the poisons they using are dangerous (TZ3_L4) |
| | | • We have been **weak** due to lack of money from mining jobs (TZ2_L2) |

Overview of selected quotations of positive and negative health outcomes, categorized by morbidity, mortality, well-being as of the coding system (quotations in the original languages can be found in the S2 Table).

contain chemicals or "*particles*", which potentially cause toxic effects, such as changing the colour of the water. According to participants, dust from coal mines was black and thick layers stacked on surfaces inside and outside the houses. Another source of dust was along the roads where heavy mining trucks pass, which was particularly perceived in sites with unpaved main access roads. In one study site in Tanzania, participants recognised that mining companies treated the main access roads occasionally to reduce the dust load, by watering the road for instance. According to participants from all countries, dust was a major issue deterring the maintenance of hygienic conditions, causing respiratory diseases and to a lesser extent also eye infections. While some of the diseases mentioned were recognised as not being new *per se*, respondents asserted increasing numbers of disease cases. Furthermore, a shift to increased chronic disease since the implementation of the mine was reported.

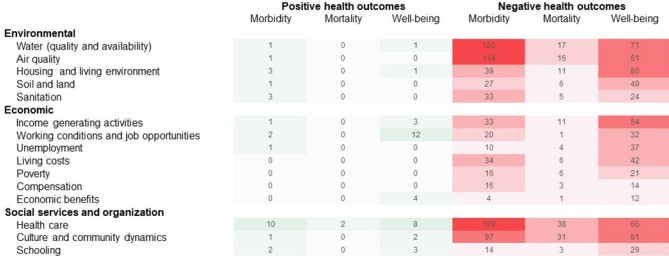

**Fig 2. Qualitative matrix of perceived impacts and health outcomes.** Distribution of perceived positive (green) and negative (red) impacts on the wider determinants of health and related health outcomes; colour gradient indicates coding frequency, numbers indicate coded references.

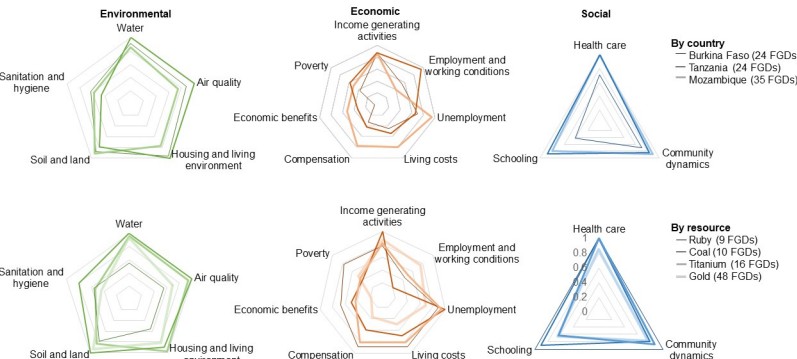

**Fig 3. Consistency of findings across countries and settings.** Radar graphs indicating proportion of focus group discussions (FGDs) where the different topics of environmental, economic and social aspects were discussed (0 not mentioned in any FGD, 1 mentioned in all FGDs).

*The whole population is suffering from cold and chronic cough. Before the [opening of the] mine toward the end of February, there was no cough and no cold. But since the mine has come these respiratory affections continue until winter. We can see the layer of dust after the blasting rising and falling on the houses, in this regard we can confirm that we are all potentially sick.*

*BF2_L8*

Bad smell was reported mostly from communities next to gold mines and was further associated with cases of fainting of students. In settings of coal, titanium and ruby extraction, dust from the mine and bad smell were mentioned considerably less often. Participants from one study site in Tanzania reported the existence of educational programs about the hazards of the bad smell and toxins around the mine, but actual health impacts were often poorly understood. Taken together, although when there were few mitigation measures in place, perceived negative health impacts associated with air pollution clearly prevailed, especially around industrial gold mines.

Themes related to people's housing and living environment were the third most important as measured by coding frequency. Participants around gold and coal mines were especially concerned about the effects related to the blasting in the mine, which caused cracks in their houses, frightening vibrations and loud noise, illustrated as follows:

*They hear the boom duuumh, tikitikitiki (the tremor due to the explosions of the mines)... And it explodes a lot, then people are scared and this causes heart problems, it provokes [affects] the heart [even] more, so we say that our health is not good.*

*MZ1_L7*

They further expressed fear of death when cracked houses collapse while families are sleeping inside. Participants in Tanzania noticed less noise when the mine was operating underground, while they were still afraid of the vibrations and cracks in the houses. Respondents from all countries perceived the increase in traffic to and away from the mine as cause of higher risk for accidents, particularly for children who are playing in the streets. However, new or improved roads were helpful for their mobility and particularly needed to access health care, as they reported.

In all countries, resettlement was a recurrent and closely interrelated theme. Some communities were hoping for resettlement to be further away from the mine and to receive compensation payment. In contrast, others felt deceived after being resettled, as the compensation was inadequate (e.g. not enough money, barren land). In Mozambique, several houses of a community were located within the mining concession and reported to be burned before the resettlement took place. Overall, these different threats, fears or insecurities affected participants' well-being, which was sometimes associated with the risk of death.

The topic of soil and land was diverse in itself and three distinct aspects were revealed. The main concerns of participants were related to decreased conditions for agriculture. They described several reasons, including the land taken or their resettlement, reduced accessibility or availability of their land, as well as soil pollution and less rainfall, which they all linked to the mining project and their activities respectively. Related to agriculture, reduced availability and poor quality of food was also mentioned. According to reports from participants, this caused hunger along with perceived stresses to provide food for their families. On the other hand, in Burkina Faso and in one study site in Tanzania, participants reported alternative farming methods ("*zaï*" in Burkina), which were introduced by mining-related companies in the case of Tanzania.

Besides agriculture, pastoralism and fishing were also impacted in similar ways. Grassland for herding became scarce with the implementation of the mine and in one study site in Mozambique access to water for fishing was restricted by the mines, as participants asserted. In one study site in Tanzania, where the "*mining dam*" (as participants referred to the tailing dam) broke, participants were upset because of the deaths of animals. According to them, the dam with the drinking water for cows was heavily polluted after the burst of the "*mining dam*", causing them to die. Thirdly, deforestation ("*cutting of forests*" in the words of the participants) was a recurrent theme among all countries and sites. Consequently, communities also lost their source of firewood for cooking as well as traditional medicinal plants, as they reported. In Burkina Faso, women struggled to collect and produce shea butter ("*beurre de karité*"), which was perceived as a particular loss, as it is not only an important ingredient of local dishes but also used for personal hygiene and businesses.

The last themes that emerged in the category of environmental impacts were related to sanitation and hygiene. Communities reported open-defecation along the mining fence, because there were no toilets installed for security guards. Inappropriate waste management was another concern. Particularly in Burkina Faso, maintaining hygienic conditions was perceived as a major challenge and often reported in relation to people's and children's health and well-being:

*It is true, we were not in perfect health before, but the problems brought by the mines made it worse. We could make efforts for the hygiene, the drinking water, our food, but with our new neighbour this is no longer possible, we cannot maintain good health. Otherwise among us, we could work together to maintain our hygiene.*

*BF2_L7*

Positively, participants appreciated interventions to improve the infrastructure such as new toilets or the donation of potties or kettles donated.

Taken together, discussion around environmental impacts was heavily negative and linked with different diseases, while improved water and road infrastructures were perceived as relief for the communities and their health.

**Economic determinants.** The discussion around economic aspects was very dynamic and the different themes that emerged were related to participants' income. Most prominently, participants reported about the impact on their traditional income generating activities, such as farming and artisanal mining. Despite having undergone reallocation, people across several communities felt that their land for agriculture or areas for artisanal mining was taken by the mine, which forced them to stop these activities, or for the latter one, continue it illegally. In two study sites in Mozambique, similar issues were also reported related to fishing, where access to the ocean was restricted for local inhabitants after the mine became operational.

On the other hand, the mines provided new job opportunities. Mostly, participants reported unskilled labour work positions, such as construction workers and security guards. Opportunities inside the mine (e.g. gardeners) were also mentioned during the discussions. However, these opportunities were reported to be very limited and related to harsh working conditions and adverse health outcomes, such as back pain and loss of sight after having worked in the mine as reported by participants from Tanzania. In Burkina Faso, announced job opportunities turned out to be empty promises. While seeking employment and regular income and, thus, being able to provide the basic needs for the family, participants were frustrated about the unfair employment conditions, harsh working conditions and social insecurity.

> I was working in the mining but later they discovered that I have sight problem and I was advised on what to do also investor was supposed to take care of me but at the end of the day he failed to do so.
>
> *TZ1_L3*

During the FGDs, unemployment was reported as a particular challenge in all countries. Therefore, they reported about their sometimes life-threatening attempts to collect stones (i.e. ore or gemstones) from the tailings of the mine for their own benefit or stealing other materials in the mine. Another strategy to generate income were sexual transactions, which were also linked to severe health consequences, including HIV. While this was reported in Mozambique and Tanzania, participants in Burkina Faso did not report on this topic. However, personal discussions of researchers and gatekeepers involved in the study indicated the same trend in Burkina Faso. Although it might particularly affect women, it is very likely that as married housewives, they did not feel comfortable talking about it.

In Mozambique and Tanzania, economic dynamics were also linked to the compensation payment of the land, when communities were resettled. However, most participants were not satisfied with this compensation or perceived it as insufficient and not adequate for what they had lost. In both countries it was reported that the received money was spent irresponsibly by men for personal amusement. Rising living costs were perceived as a challenge, especially for basic goods such as food or clean water. Particularly linked to most of the participants' poor economic situation, these impacts were related with poor health and well-being, described as stress, fatigue or weakness. Further, respondents were also concerned about not being able to afford health care, including testing and treating febrile illnesses or diarrheal diseases.

In few cases in Tanzania and Mozambique, participants also reported economic benefits for their community. For example in one study site in Tanzania, a certain share of the salaries of local security guards working for the mine were allocated for community development projects. Hence, participants felt relieved as they didn't have to contribute to the community fund. Similarly in Mozambique, participants acknowledged in-kind contributions or interventions

sponsored by the mine, as this allowed them to spend their money for personal needs, such as food and health care.

In sum, the overarching topic of having, generating or seeking income is a highly dynamic topic for communities living around industrial mines, particularly linked to the changes in local livelihoods. Although short windows of opportunities were linked to the mine, being able to afford basic needs for their health and well-being was a major concern and not always guaranteed.

**Social services & social organisation.** Perceived impacts on social services and social organisation embraced the following three heading themes: health care, community dynamics (including cultural aspects) and schooling.

In all study sites, except for one in Mozambique, interventions in the health sector sponsored by the mine led to a positive appraisal by community members. Participants mentioned different interventions, including the construction of new health care facilities, upgrading existing infrastructure, equipment and materials in health care facilities, implementation of mobile clinics or ambulance transportation, HIV testing and counselling, as well as treatment of other specific diseases. For instance, in Tanzania participants also acknowledged the support of the mines for open-cleft treatment for affected children. Despite these positively perceived health care interventions, a few participants also complained about long waiting hours and lack of services in health facilities. Further, they claimed that they had to seek health care more often due to the increased burden of disease and the loss of traditional medicinal plants.

During analysis, positive and negative issues related to internal community dynamics emerged. The introduction of the mine was associated by the participants with in-migration, which they linked to increased spread of HIV as well as HIV-related mortality. While participants from Mozambique had the same perception, they specifically stated that local people get infected when interacting with "*white people*":

> *If we were among us there in the mine and not a foreigners, we would not say that it is full of diseases [. . .] now we are mixed up with each other's disease so diseases are growing more [. . .]. Now these people from outside who come with their diseases come here and contaminate us without knowing, the reason why the diseases are increasing more and more.*
>
> *MZ3_L9*

Linked to these community dynamics, participants from all study sites perceived decreased cohesion of their family or community. For instance, young people left their family for education outside the village and men left their family with their freshly earned money or to seek other job opportunities. These dynamics were further linked to mistrust among community members. More extremely, community members felt less safe due to increased crimes among community members. For instance, a few stories of stealing were told during the FGDs. Beyond these community internal conflicts participants from one site in Mozambique anticipated the ongoing conflicts due to the natural resources. Hence, personal safety and security were particular concerns of participants, which they asserted by an increased number of deaths.

Regarding socio-cultural aspects, different spiritual conceptions and associated rituals came up during the discussions in the different countries. In Burkina Faso, different problems were linked to intrusion of mythic creatures ("*les génies et les démons*") after the implementation of the mine. Participants believed that these creatures bring bad luck and causing malformations in new-borns or even miscarriages, as illustrated in the following quotation:

*Before the mine was implemented, we had about two miscarriages per year. Nowadays, within one year we can register 20 cases of miscarriage. Women give birth to babies with malformations and we have the impression that these are geniuses and demons from the extractions, who are possessing the people and mainly the women.*

*BF1_L1*

Further, participants were worried about the future of their children. According to them, they will not have land any more, which is a traditional gift for a marriage and in turn links the current and future generations to their ancestry. Similarly to this concept reported in Burkina Faso, participants in Mozambique were not satisfied after being resettled. They were concerned about both losing the ties with the land where their ancestors were originally buried and having to bury their relatives in a new location, as "home" is where one's ancestors were born and buried. Described as concerns, worries, fear or stress during the FGDs, these changes particularly affected participant's well-being.

More positively, participants appreciated the support for local associations, interest groups or community based organisations. In Tanzania, for example, local vegetable farmers were established and beekeepers received financial support from the mine.

Another ambiguous determinant discussed, was related to education. The new construction or upgraded infrastructure in and around schools by the mining companies has contributed to improved education quality and student's attendance according to participants from all countries. On the other hand, early dropouts from school were mentioned to happen more frequently in relation to the mines in Tanzania. Participants described that this particularly affects girls when they get pregnant (from their relationships with workers from the mine). Interestingly, the topic of education was considerably less discussed in Burkina Faso, while it was mentioned in almost all discussions in Tanzania and Mozambique.

In sum, interventions to improve social services such as health care and education were common across the different settings, having potential to improve community health. On the other hand, impacts on the communities' social organisations were perceived negatively for the well-being of local populations, including various country-specific cultural aspects.

## Interlinkages between health determinants and health outcomes

While perceived impacts are reported above in distinct categories and sub-categories of the determinants of health, many impacts were closely interlinked within a specific category (e.g. water and air within the environmental determinants) or between categories (e.g. resettlement linked to environmental, social and economic determinants) as reported in this chapter.

Within the category of environmental impacts, air and water pollution were interlinked. For example, participants described the blasting as cause for air and rain water pollution at the same time. More specifically, they said that the dust released during blasting is transported through the air to their houses and then washed by the rain from the roofs directly into their water collection systems:

*Dust from the mining comes to the community and sticks to the roof of our houses, so we are not using rainy water because of the chemical carried with dust which has stacked on the roof of our houses. Despite of the contamination some people are using rainy water but they are getting health effect.*

*TZ2_L3*

Different changes induced by the mine had intersecting implications regarding the wider determinants of health. Resettlement was linked with several changes, as reported by the participants: they lost their land for agriculture (environmental impact), received compensation payment (economic impact), had to incur household goods (economic impact) or did not feel home at the new place (social impact), which ultimately affected their health and well-being. Another example is the insecure household income: due to decreased agricultural land and restrictions in the areas for artisanal mining or fishing (environmental impact), community members are hoping for employment in the mine (economic impact), while thefts and sexual transactions increased (social impact). Another example related to in-migration is illustrated as follows:

> *Native youth who were doing small mining activities in that area were enforced to shift to other distant mining areas where they can continue to conduct small mining activities. This movement caused family separation because they didn't move out with their families and as human being they will start new relationship in that area and this will accelerate to transmission of diseases meaning that if he could have been living here he wouldn't have established new relationship. But living in another community for like one year without coming back home to his wife and children it is a problem and may be when they meet one of them is already infected. I am talking about diseases like HIV because of conducting unprotected sex.*
>
> *TZ2_L4*

In light of this, only few diseases mentioned were linked to a single, specific cause. Mostly, health outcomes tended to be determined by a combination of factors, acting on different levels and changes induced by the mining project.

## Discussion

Overall, impacts of mining projects on the wider determinants of health and related health outcomes were perceived similarly by affected communities in mining areas across the study countries. An increased burden or spread of diseases was mostly related to environmental pollution and social dynamics caused by the mining operations. Elevated mortality rate was related to accidents, crimes but also to HIV and chemical exposures. With the implementation of the industrial mines, participants' perceived well-being decreased particularly through housing insecurity and changes in livelihoods. Although limited, perceived positive impacts on health and well-being were linked to upgraded infrastructure and social services such as water access points, roads, schools and health facilities, which were constructed or financially supported by the mining companies. Both positive and negative changes related to the mining projects were mostly linked to multiple determinants, and thus, closely inter-related to each other.

### Differences across countries and mining settings

Besides similarities, we observed differences of perceived health impacts due to the extracting and processing methods of resources as well as the current stage of project life cycle. For instance, gold mining caused severe environmental health impacts due to the explosive blasting and toxic chemicals used for the extraction and processing. Heavy dust emissions were also reported around coal mines. Chemicals were less discussed around titanium, ruby and coal mines, where the extraction and processing are mostly physical. However, various perceived environmental, social and economic impacts and related health implications were not only linked to extraction activities per se. As suggested by our results, which also include data

from one mining site under construction, the project implementation at large has implications for the health of surrounding communities by changes on the wider determinants. Overall, these perceived indirect health effects are also exemplified in several studies in the wider context of natural resource extraction, like poverty [35], export activities and HIV [36], health impacts of resettlement [16], nightlight and security [11]–to name a few.

Country specific aspects were primarily related to cultural beliefs and practices such as myths, rituals and traditions. Although less evident in our data, differences might also be linked to the role of the government and national regulations or laws related to the mining industry. For instance, different compensation mechanisms (e.g. providing job opportunities in Burkina Faso versus compensation payment in cash in Tanzania and Mozambique) indicated different national regulations on the local level [37]. In Tanzania, new legislative measures have been introduced in 2017, aiming to maximise the countries benefits through a "new resource nationalism", which promotes community development through mandatory investments by the mining companies [38].

## Complexity of health revealed through communities' perspective

Based on communities' perception, our study revealed various ways in which the health and well-being of local populations living and working in mining regions can be impacted. The current study provides particular insights into the complex interplay of determinants of health and health outcomes as perceived by affected communities. For instance, our findings indicate health effects linked to the environmental pollution through the mine, which further induces a shift from crop to cash based economy. Ultimately, this can lead to food insecurity as well as deprived and socially disrupted communities, with poor health conditions. Our results further show that, despite several interventions implemented by the mine, perceived negative impacts on communities' health and well-being prevailed. Drawing on this comprehensive understanding of health retrieved from affected communities expands the existing quantitative literature pertained to health risk, health determinants or health outcomes of interest. Indeed, recurrent thematic aspects reported in our study have been studied in previous research, including environmental [14, 15], social [18, 39] or economic [10, 35, 40] determinants of health. While several studies report potential health benefits [10, 41], negative aspects tend to prevail in studies synthesising health literature [18, 42, 43], which is in line with our findings. Similarly, a recent study have reported a positive trend of water infrastructure in mining regions but health benefits were less evident [44].

Studies measuring health conditions of local communities [e.g. 10, 41, 45] might have missed important aspects for the health and well-being as perceived by affected populations themselves. For example, health effects caused by cracks in the houses related to blasting and cultural dimensions of health have to our best knowledge received little attention. All these dimensions, alongside broad social engagement, need to be carefully incorporated when assessing health impacts, particularly in the dynamic setting of industrial mining projects [5, 29, 42]. A seminal study pertained to community health in mining regions monitored the health status of population in vicinity of a copper mine over time [8]. The researchers reported improvement of certain health conditions in mining communities compared to control groups, including reduced risk for malaria and less parasitic infections in children. However, this study result from a rather exceptional collaboration between the academic, health and extractive sector, which may serve as a best practice example for protecting and promoting community health in mining settings. Nonetheless, the promising findings from the comprehensive biomonitoring are in contrast with the perception of the community member participated in our study. This dissonance of quantitative measurement and observed perceptions

underlines a clear need for improved management of health impacts by acting on the wider determinants of health.

## Incorporating community perceptions in impact assessment

To address this need and prioritise community health in settings of industrial mining, there are several tools to mitigate risks and maximise benefits. For instance, impact assessment is a systematic approach to support decision-making prior project development. Impact assessment can be required by national regulations or as part of international standards or guiding principles of international organisations such as the International Finance Cooperation (IFC) or the Equator Principles [30]. At present, different types of impact assessment exist, including environmental, social, health or gender impact assessment (EIA, SIA, HIA and GIA, respectively) as well as integrated approaches, such as environmental, social and health impact assessment (ESHIA) [46–49]. HIA as stand-alone approach or integrated in other forms of impact assessment was endorsed by WHO in 1999 as holding particular promise in addressing health and its wider determinants, given its comprehensive understanding of health [50–53]. Additionally, democracy is a core value of HIA emphasising "the right of people to participate in the formulation and decision of proposals that affect their life" [54]. Placing local communities at centre and proactive inclusion of marginalised population during impact assessment may improve impact assessment by contributing context specific knowledge, which is needed for identifying and addressing communities' needs appropriately [55, 56]. Although public participation has been reported as a major challenge in impact assessment and weak methodological guidance [56, 57], community-based impact assessment approaches have been developed and successfully administered, including examples from sub-Saharan Africa [58–61].

HIA, which is poorly institutionalised in sub-Saharan African countries [30], or the inclusion of health in other forms of impact assessment should go beyond a mechanism to obtain the mining licence and also encompass participatory monitoring and follow-ups [62, 63]. As inclusive and participatory tool, HIA should inform community development initiatives including health interventions for local communities, and further contribute to sustained health improvements [8]. Meanwhile, this could strengthen Corporate Social Responsibility (CSR), which serves as umbrella term for various community development initiatives under the lead of the mining companies [64]. By suggesting appropriate interventions, HIA could ensure "real benefits" of CSR for local communities, which have been questioned [65]. Indeed, CSR holds promise to boost social and economic development of hosting countries and communities, yet more innovative solution with a broader focus on human development are needed—including the communities' perspective on their health and well-being [3, 66].

Taken together, HIA adopted by the host countries and bringing in the voices of affected communities offers a unique opportunity to "think and act" toward sustainable development, including health equity [63, 67, 68]. Beyond the global agenda for sustainable development this would also be in line with the continent's road map, aiming for increased ownership of the projects and a human-driven approach toward prosperous, integrated and peaceful Africa by 2063 [69].

## Limitations

The scope of our study is mainly limited by the following three characteristics. First, the qualitative cross-country study presents methodological, procedural as well as linguistic challenges. During data collection, a core research team was responsible for managing the fieldwork, including the training of interviewers and continuous quality control of interviews and transcription. During the initial analysis, subtle linguistic or cultural differences might have

remained concealed, as the researchers were trying to find consensus rather than cultural diversity when developing the coding system. However, particular attention to differences was paid during late stage analysis, including the discussion meetings held with the international research team.

Secondly, different types of industrial mines at different stage of the project development were selected as study sites. In Burkina Faso and Tanzania, the study was conducted around three gold mines, while in Mozambique different types of mines were selected (i.e. ruby, titanium and coal). While the choice of different types of mining industries was not very well balanced between the different countries, including different mining types allowed us to compare perceived health impacts across different settings of natural resource extraction. Additionally, only one mine was being constructed during the data collection, while all other mines have been operating since several years.

Thirdly, the qualitative study was focused on the communities' perspective, yet it would have been interesting to triangulate the communities' perspective with voices from the mine or local governments. However, systematic selection of positively and negatively impacted communities together with purposive sampling of knowledgeable community members for the FGDs should have minimised the bias of respondents' potential negative attitudes toward the mine. Additionally, the consistency of our qualitative findings from the different study sites indicates rather systematic challenges across the countries and settings.

## Conclusion

Globally, natural resources play a key role toward sustainable development, while their extraction has implications for health and well-being of local populations. Based on the perception of communities surrounding industrial mining sites in Burkina Faso, Mozambique and Tanzania, changes and health outcomes reported were consistent across the countries, whereas differences were mainly related to extraction or processing method of the resource extracted and socio-cultural aspects of health. Community perspectives revealed various impacts on the environmental, economic and social determinants of health and health outcomes related to morbidity, mortality and well-being. While negative aspects for health overweighed the positive changes, the latter were primarily related to interventions implemented by the mine. This comprehensive synthesis of perceived positive and negative impacts on health and well-being contributes to the existing literature, which is often investigating impacts on specific biomedically-oriented determinants of health or health outcomes and their quantification respectively. Importantly, community health and well-being should be addressed by acting on the wider determinants of health and including the voices of affected populations. This would further allow addressing their needs adequately, particularly those of most marginalised population groups. HIA with its potential to include and raise the voices of affected communities offers a unique opportunity for strengthening CSR related health interventions, but HIA must become adopted by hosting countries and producer regions. Compared to opportunistic health management, community-based health programs to protect and promote health and well-being in mining settings should become a priority in striving toward local, continental and global sustainable development.

## Supporting information

**S1 Fig. Outcome of participatory focus group discussions.** The picture shows the perceived impacts (written in Swahili on paper cards), which were categorised by the participants on the wider determinants of health (A4 sheets) and ultimately ranked with bottle lids.
(TIF)

**S1 Table. Excerpt from the codebook.** Thematic codes for the perceived impacts on the wider determinants of health, description of the themes and exemplary quotes extracted from the transcripts of the FGDs.
(PDF)

**S2 Table. Original quotations of coded health outcomes.** Complementary table to Table 3 with statements in original languages of selected quotations including of positive and negative health outcomes, categorized by morbidity, mortality, well-being as of the coding system.
(PDF)

**S1 Checklist. COREQ (COnsolidated criteria for REporting Qualitative research) checklist.**
(PDF)

## Acknowledgments

We are deeply grateful to all participants, sharing their time and experiences with us. We further address greatest thanks to the in-country teams, including local supervisors and study coordinators, data collectors, transcribers and drivers. We also want to acknowledge intellectual contributions of HIA4SD Project team members, other than the co-authors, who have contributed to the tool set and methodological approach employed for this study, namely Dr. Belinda Nimako. A big thank you also goes to Dr. Dominik Dietler for sharing and providing insights into his treasure trove of mining data sets and experiences from the fieldwork in Burkina Faso.

## Author Contributions

**Conceptualization:** Mirko S. Winkler, Andrea Farnham.

**Data curation:** Andrea Leuenberger, Herminio Cossa, Fadhila Kihwele, Hyacinthe R. Zabré.

**Formal analysis:** Andrea Leuenberger, Olga Cambaco, Herminio Cossa, Khátia Munguambe.

**Funding acquisition:** Mirko S. Winkler, Eusebio Macete.

**Investigation:** Olga Cambaco, Herminio Cossa, Fadhila Kihwele, Isaac Lyatuu, Hyacinthe R. Zabré, Khátia Munguambe.

**Methodology:** Andrea Leuenberger, Andrea Farnham.

**Project administration:** Mirko S. Winkler, Olga Cambaco, Isaac Lyatuu, Hyacinthe R. Zabré, Andrea Farnham.

**Supervision:** Mirko S. Winkler, Eusebio Macete, Khátia Munguambe.

**Visualization:** Andrea Leuenberger.

**Writing – original draft:** Andrea Leuenberger.

**Writing – review & editing:** Andrea Leuenberger, Mirko S. Winkler, Olga Cambaco, Herminio Cossa, Isaac Lyatuu, Hyacinthe R. Zabré, Andrea Farnham, Eusebio Macete, Khátia Munguambe.

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
