## [Decision Letter · Decision Letter 0]

4 Jan 2021

PONE-D-20-33758

Health impacts of industrial mining on surrounding communities: Local perspectives from three sub-Saharan African countries

PLOS ONE

Dear Dr. Leuenberger,

Thank you for submitting your manuscript to PLOS ONE. After careful consideration, we feel that it has merit but does not fully meet PLOS ONE’s publication criteria as it currently stands. Therefore, we invite you to submit a revised version of the manuscript that addresses the points raised during the review process.

We look forward to receiving your revised manuscript.

Kind regards,

Juliet Kiguli, MA, PhD

Academic Editor

PLOS ONE

Journal Requirements:

3. We note that Figure 1 in your submission contain map images which may be copyrighted. All PLOS content is published under the Creative Commons Attribution License (CC BY 4.0), which means that the manuscript, images, and Supporting Information files will be freely available online, and any third party is permitted to access, download, copy, distribute, and use these materials in any way, even commercially, with proper attribution. For these reasons, we cannot publish previously copyrighted maps or satellite images created using proprietary data, such as Google software (Google Maps, Street View, and Earth). For more information, see our copyright guidelines: http://journals.plos.org/plosone/s/licenses-and-copyright.

3.1.    You may seek permission from the original copyright holder of Figure 1 to publish the content specifically under the CC BY 4.0 license. 

3.2.    If you are unable to obtain permission from the original copyright holder to publish these figures under the CC BY 4.0 license or if the copyright holder’s requirements are incompatible with the CC BY 4.0 license, please either i) remove the figure or ii) supply a replacement figure that complies with the CC BY 4.0 license. Please check copyright information on all replacement figures and update the figure caption with source information. If applicable, please specify in the figure caption text when a figure is similar but not identical to the original image and is therefore for illustrative purposes only.

Reviewers' comments:

Reviewer's Responses to Questions

**Comments to the Author**

1. Is the manuscript technically sound, and do the data support the conclusions?

Reviewer #1: Yes

2. Has the statistical analysis been performed appropriately and rigorously? 

Reviewer #1: Yes

3. Have the authors made all data underlying the findings in their manuscript fully available?

Reviewer #1: Yes

4. Is the manuscript presented in an intelligible fashion and written in standard English?

Reviewer #1: Yes

5. Review Comments to the Author

Reviewer #1: Greetings,

I would like to acknowledge the authors in conducting this study. It is a very challenging research area and involves a huge planning and co-ordination which is a laborious task.

Overall the study provides an in-depth insight to the health problems and other factors determining the health among the communities around the industrial mining sites.

Abstract : Appropriately covers the content and well presented.

Main Article:

Introduction: Very elaborate and covers the required information with lacunae in existing literature also justifies the need

Methodology: The methodology could be added with more details on how the sample size was achieved? Whether the study subjects included even those who were employed in the mining industries? If so whether there was any bias in the elicited impacts? How is it taken care?

Results: Appropriate

Discussion: Elaborate and appropriate

Conclusion: The details on key determinants impacted could be included along with this sentence: ‘Community perspectives revealed various impacts on the wider determinants of health and health outcomes related to morbidity, mortality and well-being’

6. PLOS authors have the option to publish the peer review history of their article (what does this mean?). If published, this will include your full peer review and any attached files.

Reviewer #1: **Yes: **Dr. Shruthi M N

---

## [Author Response · Author response to Decision Letter 0]

26 Jan 2021

Reviewer's Responses to Questions

Comments to the Author

1. Is the manuscript technically sound, and do the data support the conclusions?

Reviewer #1: Yes

2. Has the statistical analysis been performed appropriately and rigorously?

Reviewer #1: Yes

3. Have the authors made all data underlying the findings in their manuscript fully available?

Reviewer #1: Yes

4. Is the manuscript presented in an intelligible fashion and written in standard English?

Reviewer #1: Yes

Response: We are grateful for the positive and concise appraisal of reviewer #1 regarding the questions from the editorial office.

5. Review Comments to the Author

Reviewer #1: Greetings,

I would like to acknowledge the authors in conducting this study. It is a very challenging research area and involves a huge planning and co-ordination which is a laborious task.

Overall the study provides an in-depth insight to the health problems and other factors determining the health among the communities around the industrial mining sites.

Response: Dear reviewer #1, we thank you for your positive feedback regarding our work undertaken and its relevance for the field.

Abstract : Appropriately covers the content and well presented.

Main Article:

Introduction: Very elaborate and covers the required information with lacunae in existing literature also justifies the need

Response: Thank you again, reviewer #1.

Methodology: The methodology could be added with more details on how the sample size was achieved? Whether the study subjects included even those who were employed in the mining industries? If so whether there was any bias in the elicited impacts? How is it taken care?

Response: Many thanks for mentioning this interesting issue. The recruitment process was indeed complex and we have further elaborated the section in the manuscript. To state more clearly how the sample size was achieved, we have added the sentence that we planned to conduct two to four session with 6 - 10 participants in the communities identified, to read: In order to generate comparative data from the different settings and countries, two to four sessions were planned with 6 -10 participants per session (see revised manuscript lines 198-200).

Further, we have specified in the description of the study population that only few participants (n = 13) were employed by the mines at the time point of data collection. It reads now as follows: Only few participants reported to be employed by the mining companies (n = 13) (see revised manuscript line 260). Notably, being employed or not was not a recruitment criterion to participate in the study. But the low number may reflects the restricted number of job opportunities that were mentioned by the participants. Against this background, this might have contributed to their rather negative attitude, which we point out to in the limitations. At the same time, we also like to emphasize that the consistency of the findings, including the low number of community members employed by the mines, indicates a systematic problem across the settings and countries. We have added this reflection to the limitations, to read: Additionally, the consistency of our qualitative findings from the different study sites indicates rather systematic challenges across the countries and settings (see revised manuscript lines 708-709).

Results: Appropriate

Discussion: Elaborate and appropriate

Response: Thank you, reviewer #1 for considering the results and discussion section as appropriate.

Conclusion: The details on key determinants impacted could be included along with this sentence: ‘Community perspectives revealed various impacts on the wider determinants of health and health outcomes related to morbidity, mortality and well-being’

Response: Thank your brining up this point. To be more specific about the wider determinants of health, we have specified the phrase to “environmental, economic and social determinants of health” (see manuscript line 717).

6. PLOS authors have the option to publish the peer review history of their article (what does this mean?). If published, this will include your full peer review and any attached files.

Do you want your identity to be public for this peer review? For information about this choice, including consent withdrawal, please see our Privacy Policy.

Reviewer #1: Yes: Dr. Shruthi M N

Response: We are delighted to see that reviewer #1 is willing to make the review available for the public. Thank you, Dr. Shruthi, for all your efforts on our piece!

---

## [Decision Letter · Decision Letter 1]

21 Apr 2021

PONE-D-20-33758R1

Health impacts of industrial mining on surrounding communities: local perspectives from three sub-Saharan African countries

PLOS ONE

Dear Dr. Leuenberger,

Thank you for submitting your manuscript to PLOS ONE. After careful consideration, we feel that it has merit but does not fully meet PLOS ONE’s publication criteria as it currently stands. Therefore, we invite you to submit a revised version of the manuscript that addresses the points raised during the review process especially the review #-2. It is a very important and complex area of investigation in the field of environmental and occupational health and safety. 

Please submit your revised manuscript by Jun 05 2021 11:59PM. I look forward to your revised manuscript, and hope to have an expeditious decision.  If you will need more time than this to complete your revisions, please reply to this message or contact the journal office at plosone@plos.org. Please include the following items when submitting your revised manuscript:

We look forward to receiving your revised manuscript.

Kind regards,

Alok Deoraj, PhD

Academic Editor

PLOS ONE

Reviewers' comments:

Reviewer's Responses to Questions

**Comments to the Author**

1. If the authors have adequately addressed your comments raised in a previous round of review and you feel that this manuscript is now acceptable for publication, you may indicate that here to bypass the “Comments to the Author” section, enter your conflict of interest statement in the “Confidential to Editor” section, and submit your "Accept" recommendation.

Reviewer #2: (No Response)

2. Is the manuscript technically sound, and do the data support the conclusions?

Reviewer #2: Yes

3. Has the statistical analysis been performed appropriately and rigorously? 

Reviewer #2: Yes

4. Have the authors made all data underlying the findings in their manuscript fully available?

Reviewer #2: No

5. Is the manuscript presented in an intelligible fashion and written in standard English?

Reviewer #2: No

6. Review Comments to the Author

Reviewer #2: PLOS ONE Review (PONE-D-20-33758)

Health impacts of industrial mining on surrounding communities: Local perspectives from three sub-Saharan African countries.

The manuscript is tackling very important public health issues regarding impact of mining on the surrounding populations in Burkina Faso, Mozambique, and Tanzania. The design of the study was well done.

However, there are a number of missing information that may be instrumental in addressing these issues.

First of all, the manuscript needs to be reviewed by a native English speaker or someone with excellent command of English.

The missing information (that can be easily retrieved and hence enhance the quality of the manuscript) is noted in the following lines:

1. What is the role of the governments and the mining companies in this context?

2. Are there any environmental laws and regulations in these countries dealing with mining?

3. Do the three countries have ministries of labour, mining, environment, …? If we are to design preventive measures, there has to a government’s agency to go to in order to come up with ways to control/prevent the negative impact of mining on the surrounding populations.

4. What ILO, or WHO, … says about these issues? Any recommendations from these United Nations’ agencies, or from any international organization.

5. On Page 17, under “Environmental determinants”, what is the role of the governments regarding the reported impacts related to water, air, housing and living environment, soil and land and sanitation and hygiene?

6. On Page 21, under “Economic determinants”, could the reader know the position of the governments regarding displaced populations? What do laws (if they exist) say about this?

7. On Page 23, under “Social services and social organization”, if these governments and/or mining companies secured lending, they should abide by the World Bank IFC (International Finance Corporation) Equator principles* and performance standards. *These are a set of voluntary social and environmental principles which must be met to satisfy the conditions of lending.

8. On Page 25, under “Interlinkages between health determinants and health outcomes”, are any policies from the mining companies that deal with these?

7. PLOS authors have the option to publish the peer review history of their article (what does this mean?). If published, this will include your full peer review and any attached files.

Reviewer #2: No

---

## [Author Response · Author response to Decision Letter 1]

12 May 2021

Dear Dr. Deoraj

We were delighted to receive your e-mail dated 21 April 2021, which provided encouraging feedback and constructive comments and suggestions from one external reviewer in connection with the aforementioned manuscript. After a first round of external peer-review, we were actually slightly surprised to receive an additional reviewer report that seems unrelated to the first revision. Yet, we revised our piece once more in light of the reviewer’s comments and tried our level best to address the concerns raised.

Below, please find our point-by-point response, clearly indicating how and where in the manuscript changes have been made (line numbers). In order to readily assist you in tracking our changes, they have been highlighted using a yellow marker. We also enclose a “clean” version of the revised manuscript as supporting information.

We very much hope that our revised manuscript is now suitable for publication for PLOS ONE and are looking forward to hearing from you. 

Yours sincerely,

Andrea Leuenberger on behalf of all authors

Reviewers' comments:

Reviewer's Responses to Questions

Comments to the Author

1. If the authors have adequately addressed your comments raised in a previous round of review and you feel that this manuscript is now acceptable for publication, you may indicate that here to bypass the “Comments to the Author” section, enter your conflict of interest statement in the “Confidential to Editor” section, and submit your "Accept" recommendation.

Reviewer #2: (No Response)

2. Is the manuscript technically sound, and do the data support the conclusions?

Reviewer #2: Yes

3. Has the statistical analysis been performed appropriately and rigorously? 

Reviewer #2: Yes

4. Have the authors made all data underlying the findings in their manuscript fully available?

Reviewer #2: No

5. Is the manuscript presented in an intelligible fashion and written in standard English?

Reviewer #2: No

Response: We are grateful for the overall positive and concise appraisal of reviewer #2 regarding the questions from the editorial office and have noted that the language is a particular concern. Therefore, one co-author, who is a native English speaker, focused on language edits specifically.

6. Review Comments to the Author

Reviewer #2: PLOS ONE Review (PONE-D-20-33758)

Health impacts of industrial mining on surrounding communities: Local perspectives from three sub-Saharan African countries.

The manuscript is tackling very important public health issues regarding impact of mining on the surrounding populations in Burkina Faso, Mozambique, and Tanzania. The design of the study was well done.

However, there are a number of missing information that may be instrumental in addressing these issues.

Response: Dear reviewer #2, we thank you for your positive feedback on the research undertaken and considering it a relevant contribution to the scientific literature. In light of your comments, we have further revised our paper based on the communities’ perspective in a concise manner as indicated below.

First of all, the manuscript needs to be reviewed by a native English speaker or someone with excellent command of English.

Response: We addressed the comments related to the content of the manuscript and paid particular attention to grammar and syntax. As mentioned above, one co-author, who is a native English speaker, focused on language edits specifically.

The missing information (that can be easily retrieved and hence enhance the quality of the manuscript) is noted in the following lines:

1. What is the role of the governments and the mining companies in this context?

2. Are there any environmental laws and regulations in these countries dealing with mining?

Response: We thank reviewer #2 for bringing up the role of governments and national laws or regulations in relation to industrial mining projects. To address the comment, we have integrated a sentence in the description of the study sites, to read: “In all study countries, environmental impact assessments (EIA) are required by law to obtain a mining license. Yet, specific national regulations for health impact assessment and the rigorous inclusion of health in other forms of impact assessment remain an unmet need [Winkler et al., 2013, Dietler et al., 2020]” (see revised manuscript lines 116-120).

Based on our data, the role of the governments as well as the differences in governancebetween the countries were less evident. Only a few participants mentioned the government or national regulations in relation to perceived health impacts. Despite the fact that governmental issues were identified as secondary themes in our study, the need to incorporate health impacts in national regulations and the implementation thereof is discussed in last section of the discussion, (“Incorporating community perceptions in impact assessment”, see revised manuscript lines 652-691).

3. Do the three countries have ministries of labour, mining, environment, …? If we are to design preventive measures, there has to a government’s agency to go to in order to come up with ways to control/prevent the negative impact of mining on the surrounding populations.

Response: This is very valid point and relates to the overarching aim of the large research initiative (HIA4SD Project) behind this paper, which is to promote the use and implementation of updated guidelines regarding health impact assessment (HIA) in the respective countries. The initiative collaborates with national stakeholders, including representatives from the public and private sector, such as ministries of mines and health. The current piece is, however, designed to focus on the voice of affected communities. While we acknowledge the importance of the public sector for national health strategies, it is beyond the scope of this manuscript to describe the different health policies and practices of the different countries.

4. What ILO, or WHO, … says about these issues? Any recommendations from these United Nations’ agencies, or from any international organization.

Response: In the past decades, responsible business conduct has gained momentum in the extractive industry sector. For instance, the “United Nations Guiding Principles on Business and Human Rights” are a key international document to protect and respect the human rights. This can also include impact assessment practices, which are required either on a legal basis or by international organisations. We have specified this in the manuscript by adding the following phrases: “Impact assessment can be required by national regulations or as part of international standards or guiding principles of international organisations such as the International Finance Cooperation (IFC) or the Equator Principles [Winkler et al., 2013]. At present, different types of impact assessment exist, including environmental, social, health or gender impact assessment (EIA, SIA, HIA and GIA, respectively) as well as integrated approaches, such as environmental, social and health impact assessment (ESHIA) [Esteves et al., 2012; Harris-Roxas et al., 2012; Morgan, 2012; Hill et al., 2017]. HIA as a stand-alone approach or integrated in other forms of impact assessment was endorsed by WHO in 1999 as holding particular promise in addressing health and its wider determinants, given its comprehensive understanding of health [Leuenberger et al., 2019; Winkler et al., 2020; Winkler et al., 2021] ” (see revised manuscript lines 657-665). 

 This paper reflects the perspective of communities affected by industrial mining as part of a larger research initiative aiming to strengthen HIA in sub-Saharan Africa. As this manuscript focuses on perspectives from the local level, opinions from other local, national or international stakeholders were not integrated. This is, however, done under the ‘governance research stream’ of the HIA4SD Project.

5. On Page 17, under “Environmental determinants”, what is the role of the governments regarding the reported impacts related to water, air, housing and living environment, soil and land and sanitation and hygiene?

Response: The “environmental determinants” section reflects the communities’ perspectives of the impacts on the wider determinants of health, and as such the content is driven by the participants themselves. While reporting on perceived impacts on the environmental determinants of health, participants did not specifically mention the role of the government. During the analysis, statements related to the role of the government were categorized as social impacts. While investigating the (perceived) role of the government would be highly interesting, we prefer to report the communities’ perspectives in a unbiased and impartial manner. 

In addition to the current qualitative approach, we would like to mention the quantitative and mixed methods research pertaining to environmental health impacts, which is also associated with the framing research project and have been published recently: Dietler et al., 2021, Leuenberger and Dietler et al., 2021.

6. On Page 21, under “Economic determinants”, could the reader know the position of the governments regarding displaced populations? What do laws (if they exist) say about this?

Response: Many thanks for pointing out role of the governments and laws in relation to displaced populations. As above, this is also part of the results section reporting the impacts as perceived by affected communities. While governance aspects are covered within the larger research initiative framing the manuscript at hand, an analysis of the national laws is beyond the scope of this manuscript. 

7. On Page 23, under “Social services and social organization”, if these governments and/or mining companies secured lending, they should abide by the World Bank IFC (International Finance Corporation) Equator principles* and performance standards. *These are a set of voluntary social and environmental principles which must be met to satisfy the conditions of lending.

Response: Thank you bringing up this important point. Indeed, international standards and principles are an important measure to ensure the well-being of affected communities. Our research group has produced several publications that reflect on these international performance standards and how they influence HIA practice (e.g. Winkler et al., 2013; Winkler et al, 2020). Hence, we are very familiar with the point raised by the reviewer here and fully concur on their importance. However, the present qualitative paper purely reflects the communities’ perspectives, which apparently seem to be in contrast with international standard principles and standards. The main objective of the paper is to raise the voice of the communities and not to judge the performance of mining companies or governments. Based on our data obtained from affected communities, we would not feel comfortable to judge the governments’ or mining companies’ performance on environmental and social indicators and hence, prefer not to discuss the justification of lending processes.

8. On Page 25, under “Interlinkages between health determinants and health outcomes”, are any policies from the mining companies that deal with these?

Response: This section is part of the results and reports about the community’s perspective. During the focus group discussion, participants did not mention any policies from the mining companies to address the complexity of health impacts, meaning policies acting on health policies to ultimately improve health and well-being. In order to answer to the question raised by the reviewer, we would like to refer to the Good Practice Guidance on Health Impact Assessment (2010), which was issued by the International Council on Minerals and Metals (ICMM) in 2010. This guidance adopted by many mining projects specifically addressed the interlinkages between health determinants and health outcomes.

7. PLOS authors have the option to publish the peer review history of their article (what does this mean?). If published, this will include your full peer review and any attached files.

Do you want your identity to be public for this peer review? For information about this choice, including consent withdrawal, please see our Privacy Policy.

Reviewer #2: No

References added: 

Dietler D, Lewinski R, Azevedo S, Engebretsen R, Brugger F, Utzinger J, et al. Inclusion of health in impact assessment: a review of current practice in sub-Saharan Africa. International Journal of Environmental Research and Public Health. 2020;17(11):4155.

WHO European Centre for Health Policy. Gothenburg Consensus Paper. Brussels, Belgium: World Health Organization Regional Office for Europe; 1999.

Winkler MS, Viliani F, Knoblauch AM, Cave B, Divall M, Ramesh G, et al. Health impact assessment international best practice principles. Fargo, USA: International Association for Impact Assessment; 2021.

---

## [Decision Letter · Decision Letter 2]

17 May 2021

Health impacts of industrial mining on surrounding communities: local perspectives from three sub-Saharan African countries

PONE-D-20-33758R2

Dear Dr. Leuenberger,

We’re pleased to inform you that your manuscript has been judged scientifically suitable for publication and will be formally accepted for publication once it meets all outstanding technical requirements.

Kind regards,

Alok Deoraj, PhD

Academic Editor

PLOS ONE

Additional Editor Comments (optional):

Congratulations, Dr. Leuenberger and your team! This is an important work your team is doing which is directly related to SDGs. Thank you for your relevant work. 

Reviewers' comments:

Reviewer's Responses to Questions

**Comments to the Author**

1. If the authors have adequately addressed your comments raised in a previous round of review and you feel that this manuscript is now acceptable for publication, you may indicate that here to bypass the “Comments to the Author” section, enter your conflict of interest statement in the “Confidential to Editor” section, and submit your "Accept" recommendation.

Reviewer #1: All comments have been addressed

Reviewer #2: All comments have been addressed

2. Is the manuscript technically sound, and do the data support the conclusions?

Reviewer #1: Yes

Reviewer #2: Yes

3. Has the statistical analysis been performed appropriately and rigorously? 

Reviewer #1: Yes

Reviewer #2: Yes

4. Have the authors made all data underlying the findings in their manuscript fully available?

Reviewer #1: Yes

Reviewer #2: Yes

5. Is the manuscript presented in an intelligible fashion and written in standard English?

Reviewer #1: Yes

Reviewer #2: Yes

6. Review Comments to the Author

Reviewer #1: (No Response)

Reviewer #2: I am quite satisfied with the revisions to the manuscript. Indeed, you addressed all the concerns and questions that I had. Thanks

7. PLOS authors have the option to publish the peer review history of their article (what does this mean?). If published, this will include your full peer review and any attached files.

Reviewer #1: **Yes: **Dr. Shruthi M N

Reviewer #2: No

---

## [Editor Report · Acceptance letter]

26 May 2021

PONE-D-20-33758R2 

Health impacts of industrial mining on surrounding communities: local perspectives from three sub-Saharan African countries 

Dear Dr. Leuenberger:

I'm pleased to inform you that your manuscript has been deemed suitable for publication in PLOS ONE. Congratulations! Your manuscript is now with our production department. 

Kind regards, 

on behalf of

Dr. Alok Deoraj 

Academic Editor

PLOS ONE